# SSDD-GAN: Single-Step Denoising Diffusion GAN for Cochlear Implant Surgical Scene Completion

**Yike Zhang**[1]                                                          YIKE.ZHANG@VANDERBILT.EDU

**Eduardo Davalos**[2]                        EDUARDO.DAVALOS.ANAYA@VANDERBILT.EDU

**Jack Noble**[3]                                                      JACK.NOBLE@VANDERBILT.EDU

## Abstract

Recent deep learning-based image completion methods, including both inpainting and out-painting, have demonstrated promising results in restoring corrupted images by effectively filling various missing regions. Among these, Generative Adversarial Networks (GANs) and Denoising Diffusion Probabilistic Models (DDPMs) have been employed as key generative image completion approaches, excelling in the field of generating high-quality restorations with reduced artifacts and improved fine details. In previous work, we developed a method aimed at synthesizing views from novel microscope positions for mastoidectomy surgeries; however, that approach did not have the ability to restore the surrounding surgical scene environment. In this paper, we propose an efficient method to complete the surgical scene of the synthetic postmastoidectomy dataset. Our approach leverages self-supervised learning on real surgical datasets to train a Single-Step Denoising Diffusion-GAN (SSDD-GAN), combining the advantages of diffusion models with the adversarial optimization of GANs for improved Structural Similarity results of 6%. The trained model is then directly applied to the synthetic postmastoidectomy dataset using a zero-shot approach, enabling the generation of realistic and complete surgical scenes without the need for explicit ground-truth labels from the synthetic postmastoidectomy dataset. This method addresses key limitations in previous work, offering a novel pathway for full surgical microscopy scene completion and enhancing the usability of the synthetic postmastoidectomy dataset in surgical preoperative planning and intraoperative navigation.

**Keywords:** Image completion, image inpainting, image outpainting, image synthesis, surgical scene synthesis, postmastoidectomy, cochlear implant surgery, Denoising Diffusion Probabilistic Models (DDPMs), Generative Adversarial Networks (GANs), Diffusion-GAN.

## 1. Introduction

Cochlear Implant (CI) procedures are transformative surgeries that aim to restore hearing for individuals with moderate-to-profound hearing disabilities, offering a way to improve communication and quality of life (Labadie and Noble, 2018). These procedures involve the precise placement of an electrode array into the cochlea, enabling direct stimulation of the auditory nerve to restore hearing ability (Zhang and Noble, 2023; Zhang et al., 2024a). As one of the initial steps in CI surgery, mastoidectomy involves the careful removal of portions of the temporal bone to create access to the middle ear and cochlea. This procedure ensures a clear pathway for electrode array insertion while safeguarding critical anatomical structures, such as the facial nerve and the chorda. We hypothesize that if the surgically created mastoidectomy surface can be predicted directly from preoperative CT scans, it could

serves as a valuable resource for numerous downstream tasks, including surgical tool tracking, surgical scene synthesis, and pose estimation of anatomical structures. These potential benefits could collectively contribute to improved surgical navigation and enhanced intraoperative visualization, ultimately supporting greater precision and optimizing the placement of the electrode array during cochlear implantation. Recent studies have increasingly focused on leveraging advanced imaging and deep learning-based methods to assist surgeons in understanding and navigating complex anatomical structures during cochlear implant surgery. In our previous work (Zhang et al., 2024b; Zhang and Noble, 2024), we introduced novel methodologies for reconstructing postmastoidectomy surfaces and synthesizing novel views from a single microscopy image, as the pipeline shown in Figure 1. The synthesized postmastoidectomy scenes are generated by directly generate new camera poses to produce multiple viewpoints. These methods demonstrated significant potential in providing partial reconstructions of the surgical scene, improving intraoperative visualization, and eliminating reliance on external tracking devices. However, these approaches were limited to texturing the postmastoidectomy surface from preoperative CT scans, neglecting the broader surgical environment captured by the microscopy. The absence of contextual information surround-

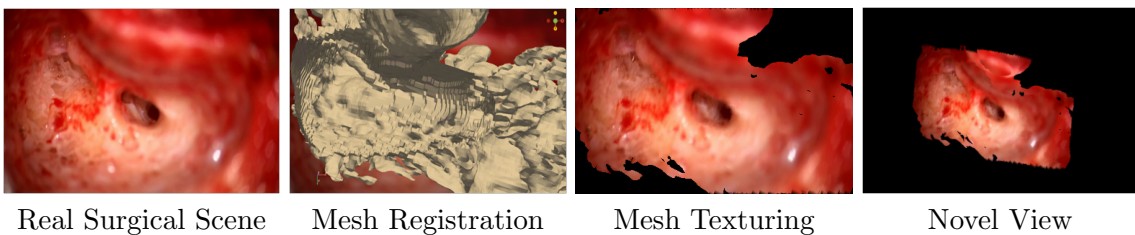

Real Surgical Scene    Mesh Registration    Mesh Texturing    Novel View

Figure 1: **Synthetic Postmastoidectomy Surgical Dataset Generation.** Pipeline for synthesizing a postmastoidectomy surgical scene.

ing the surgical site poses challenges for comprehensive scene understanding, particularly in scenarios where broader spatial awareness is critical for decision-making. To address these limitations, this paper proposes a novel approach that leverages image completion using a deep learning-based generative model to fill in the missing regions of the surgical scene, enabling the synthesis of a complete and detailed surgical environment. Image completion involves reconstructing missing or occluded regions (inpainting) or extending an image beyond its boundaries (outpainting) by leveraging contextual information from the known areas. It is a critical task in computer vision with wide-ranging applications in photo editing, image-based rendering, and computational photography (Park et al., 2017; Sabini and Rusak, 2018; Suvorov et al., 2021). The primary challenge lies in generating visually realistic and semantically meaningful pixels for the missing regions while ensuring seamless coherence with the known content.

Recent popular works in image inpainting and outpainting are heavily based on deep learning neural networks. In the research proposed by (Pathak et al., 2016), a context encoder was introduced to predict missing image regions using convolutional neural networks (CNNs), laying the foundation for generative approaches to image inpainting. Subsequent advancements, such as those by (Iizuka et al., 2017), incorporated both global and local

discriminators to enhance texture consistency, while (Yu et al., 2018) introduced the use of contextual attention mechanisms for more realistic inpainting of irregular holes. These developments have significantly improved the quality and applicability of image completion techniques across various domains. Surgical data often contains complex anatomical structures, cluttered scenes, and occlusions caused by surgical tools or the surgeon's hands. Beyond visual realism, inpainting for surgical scenes requires clinically meaningful reconstructions with high geometric fidelity to preserve critical anatomical details. These challenges require novel and effective approaches that can adapt to the complexities of real surgical scenes without relying heavily on manual annotations. For instance, (Daher et al., 2023) introduced a machine learning approach using a temporal generative adversarial network (GAN) to inpaint hidden anatomy under specularities. This paper aims to reconstruct the complete post-mastoidectomy surgical scene by training a neural network on real surgical datasets. Additionally, it focuses on building a patient-specific dataset to assist intraoperative registration between preoperative CT scans and the corresponding surgical scene. With the proposed self-supervised Single-Step Denoising Diffusion-GAN (SSDD-GAN) framework, our approach bypasses the need for manual annotations by learning directly from the inherent structures in the data, enabling accurate and clinically relevant surgical scene synthesis. Our contributions can be summarized in the following:

- **Novel Self-supervised Image Completion Framework SSDD-GAN:** We introduce an image completion framework SSDD-GAN that aims for surgical scene inpainting and outpainting. This self-supervised approach eliminates the need for manually annotated datasets, ensuring training efficiency and generalizability.

- **Zero-shot Synthesis using the Synthetic Postmastoidectomy Dataset:** Our goal is to generate a complete surgical scene by utilizing the synthetic postmastoidectomy dataset via a zero-shot learning strategy by training and validating the model on real surgical datasets.

- **Enhanced Cochlear Implant Surgery Visualization and Navigation:** The proposed method provides full surgical field visualizations along with precise camera pose information derived from the previously synthetic postmastoidectomy dataset. This advancement paves the way for surgical scene understanding, tool tracking, and anatomical navigation, offering the potential for improving cochlear implant surgery preoperative planning and intraoperative guidance.

## 2. Methodology

To address the limitations outlined in (Zhang and Noble, 2024) and enable the completion of a surgical scene, we propose a deep-learning-based approach trained and validated on a dataset of real microscopy views. Given the irregular shapes of the synthetic post-mastoidectomy surgical views (shown in the first row of Figure. 7), we generate random masks on the real surgical dataset to simulate the partially generated postmastoidectomy multi-views. This dataset creation strategy ensures that the model effectively learns to restore missing regions while maintaining robustness to the diverse shapes and irregularities of the synthetic postmastoidectomy scenes. To simulate the partially generated surgical

scenes using the postmastoidectomy surface, we automatically generate masks on the surgical frames using a range of polygonal shapes containing randomly placed holes. This label-generation approach effectively mimics the irregularities and variability observed in the synthetic post-mastoidectomy scene dataset. We propose SSDD-GAN that combines the strengths of diffusion models and GANs to synthesize realistic surgical scenes guided by randomly masked real surgical data. While traditional diffusion models (DDPMs) (Ho et al., 2020) have noticeable advantages in generating synthetic images, audio, and videos, they often suffer from slow inference times due to their long iterative sampling process. This limitation also presents challenges when attempting to integrate a discriminator into the denoising routine. Unlike traditional DDPMs, which rely on the iterative denoising process, our method focuses exclusively on single-step denoising and reconstruction to minimize computational cost by directly mapping noise to data. In general, diffusion models can be viewed as a special type of variational autoencoders (VAEs) (Sohl-Dickstein et al., 2015). Different from VAE-based models, our method maintains the diffusion formulation by training with progressive noise adding and subsequent denoising routine. The proposed framework introduce controlled noise to the input and directly learning a single-step denoising operation. This feature preserves the diffusion-inspired noise-and-denoise training objective, though simplified into just one denoising step at inference. Moreover, this structure also benefits from the adversarial training provided by a GAN discriminator. This combination allows SSDD-GAN to efficiently produce high-quality image reconstructions with faster sampling speed. Adding a discriminator to the denoising routine can lead to promising results since the discriminator provides an additional adversarial component that helps the diffusion model output better results (Wang et al., 2023). Leveraging these advantages, we aim to enhance sampling efficiency and further improving the quality and realism of the completed surgical scenes. The forward diffusion process of our method is shown in Figure 2. As shown in the figure, we only apply the Gaussian noise on the non-masked region in the forward diffusion process. The data points for the forward diffusion process are

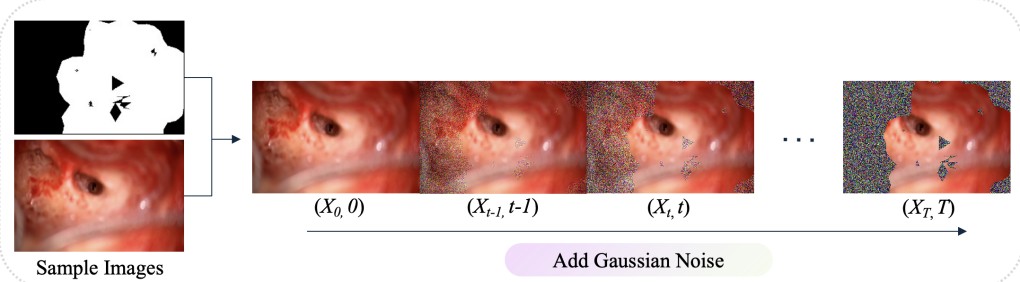

Figure 2: **Forward Diffusion Process**. We preserve the masked region of the original sample data while applying Gaussian noise exclusively to the non-masked region.

sampled from a real data distribution $x_t \sim q(x)$. This process progressively adds Gaussian noise to the targeted region (black pixels) in the samples over $T$ steps, where $T \in [700, 900]$, as determined by our experiments. We produce a sequence of noisy samples $x_1, ..., x_T$. The interval sizes are controlled by a linear beta scheduler $\{\beta_t \in (0,1)\}_{t=1}^{T}$. The whole standard

forward diffusion process can be expressed in the following Eq 1:

$$q(x_t|x_{t-1}) = \mathcal{N}(x_t; \sqrt{1-\beta_t}x_{t-1}, \beta_t I),$$
$$q(x_{1:T}|x_0) = \prod_{t=1}^{T} q(x_t|x_{t-1}) \tag{1}$$

At any arbitrary time step $t$, we can sample $x_t$ in a closed form using the re-parametrization method, which the method can be described as the following Eq 2. We set the random variable $z$, $q_\phi(z|x)$ as a multivariate Gaussian, and $\epsilon$ is an auxiliary independent random variable.

$$z \sim q_\phi(z|x^i) = \mathcal{N}(z; \mu^i, \sigma^{2(i)}I),$$
$$z = \mu + \sigma \odot \epsilon, \text{ where } \epsilon \sim \mathcal{N}(0, I) \tag{2}$$

Gaussian noise can be directly added from $x_0$ to any arbitrary step $x_t$ using the following Eq 3. Let $\alpha_t = 1 - \beta_t$, $\beta_t = 1 - \alpha_t$, $\bar{\alpha}_t = \prod_{i=1}^{t} \alpha_i$, and $\delta$ denotes for the generated mask regions:

$$x_t = (\sqrt{\alpha_t}x_{t-1} + \sqrt{1-\alpha_t}\epsilon_{t-1})(1-\delta) + \delta x_0, \quad \text{where } \epsilon_{t-1} \sim \mathcal{N}(0, I),$$
$$= (\sqrt{\alpha_t \alpha_{t-1}}x_{t-2} + \sqrt{1-\alpha_t \alpha_{t-1}}\epsilon_{t-1}\epsilon_{t-2})(1-\delta) + \delta x_0,$$
$$= ...$$
$$= (\sqrt{\bar{\alpha}_t}x_0 + \sqrt{1-\bar{\alpha}_t}\bar{\epsilon})(1-\delta) + \delta x_0, \quad \text{where } \bar{\epsilon} \text{ merges } \epsilon_{t-1}, \epsilon_{t-2}, ... \text{ Gaussians.} \tag{3}$$

The term $(\sqrt{\bar{\alpha}_t}x_0 + \sqrt{1-\bar{\alpha}_t}\bar{\epsilon})(1-\delta)$ represents a partial forward-diffusion mix of the clean image $x_0$ and merged Gaussian noise $\bar{\epsilon}$, scaled by $(1-\delta)$. The term $\delta x_0$ adds back a fraction $\delta$ of the original image $x_0$ to $x_t$. We progressively reduce the signal in the masked region of the original image sample $x_0$ by a factor of $\sqrt{\bar{\alpha}_t}$, while simultaneously adding noise to the masked region scaled by $\sqrt{1-\bar{\alpha}_t}\bar{\epsilon}$. The proposed single-step denoising process is shown in Figure 3. $\delta x_0$ denotes the black region in the inverted mask. The denoising U-Net structure is adopted from the method proposed in (Ho et al., 2020). Unlike the iterative denoising process proposed in their method, our approach directly employs a neural network to predict the noise $\epsilon_t$ and map any arbitrary step $x_t$ to a reconstruction of $x_0$ in a single step.

$$x_0 = \left(\frac{x_t - \sqrt{1-\bar{\alpha}_t}\bar{\epsilon}}{\sqrt{\bar{\alpha}_t}}\right)(1-\delta) + \delta x_t \tag{4}$$

$\delta x_t$ refers to masked regions that are identical to the corresponding regions in the original image $\delta x_0$ throughout the diffusion process. During training, we use Mean Squared Error (MSE) loss for noise prediction. The reconstructed images are then compared with their corresponding sample data using Structural Similarity Index (SSIM) loss, further refining the model and improving the reconstruction quality. This direct denoising method significantly reduces computational time while enabling the integration of a discriminator for enhanced performance. Specifically, we implement a Patch-GAN discriminator(Isola et al., 2018), which evaluates image structure at the patch level. The discriminator classifies whether each $N$ by $N$ patch in an image is real or fake by applying a convolutional filter across the entire image and gathering the responses to produce the final output. This approach ensures that the model focuses on local details while maintaining computational efficiency. The discriminator is trained using the BCEWithLogits loss function that focuses solely on inputs from the generated content region and the corresponding real surgical scene region.

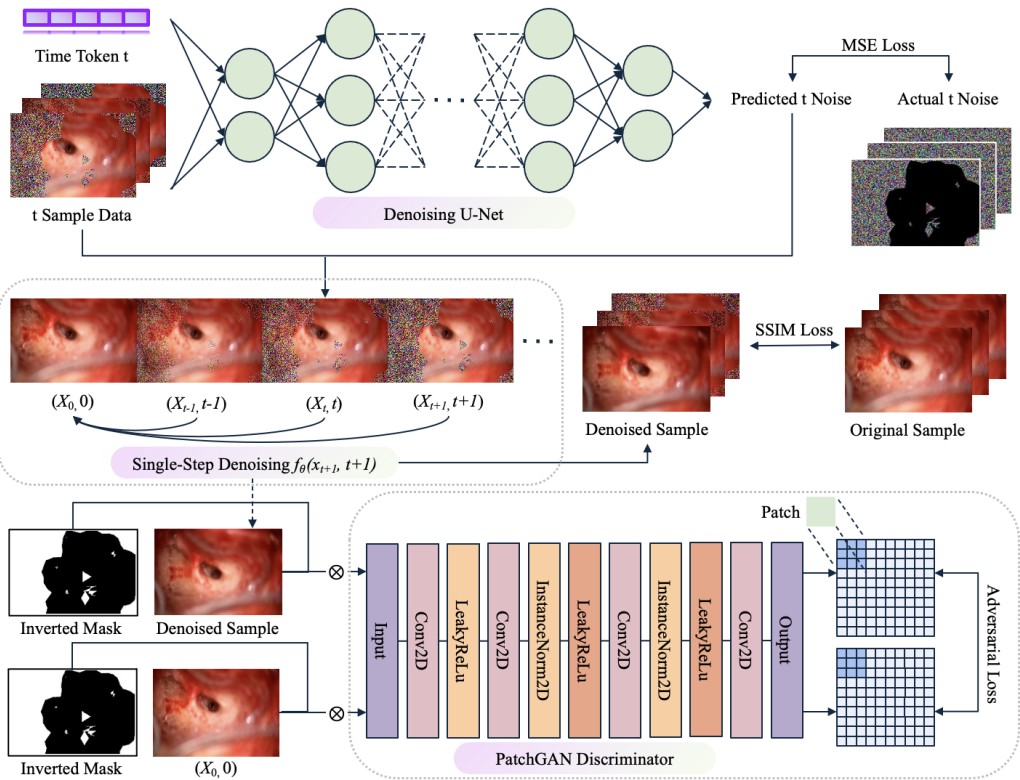

Figure 3: **Single-Step Denoising Diffusion Process**. We incorporate a discriminator in this process to further improve the realism of synthetic samples.

## 3. Results

### 3.1. Performance Evaluation

Our dataset comprises 932 real surgical frames collected from a cochlear implant surgery on a patient, divided into training, validation, and testing sets in a ratio of 0.75, 0.15, and 0.15, respectively. The quantitative results are summarized in Table 1, comparing our method to other generative models, such as CycleGAN(Zhu et al., 2020), Pix2Pix(Isola et al., 2018), DeepFillv2(Yu et al., 2019), and PEIPNet(Ko et al., 2023). We use metrics such as Fréchet Inception Distance (FID)(Heusel et al., 2018), Kernel Inception Distance (KID)(Bińkowski et al., 2021), Learned Perceptual Image Patch Similarity (LPIPS)(Zhang et al., 2018), Inception Score (IS)(Salimans et al., 2016), Peak Signal-to-Noise Ratio (PSNR), and Structural Similarity Index (SSIM)(Wang et al., 2004) to measure the overall performance numerically. The results in Table 1 show that the proposed method outperforms other methods in most metrics. Figure 4 provides a detailed comparison of the aforementioned methods, evaluating their performance using L1(%), PSNR, and SSIM metrics. These metrics collectively assess the accuracy, reconstruction quality, and structural consistency of each method. From Figure 4(a-c), we observe that our proposed method consistently outperforms competing models across all evaluated thresholds, demonstrating its robustness and superior recon-

| Methods | FID ↓ | KID ↓ | LPIPS ↓ | L1(%) ↓ | PSNR ↑ | SSIM ↑ |
|---|---|---|---|---|---|---|
| CycleGAN | 0.612 | 0.131 | 0.262 | 9.441 | 17.058 | 0.598 |
| PEIPNet | 1.222 | 0.087 | 0.149 | 3.965 | 24.584 | 0.763 |
| Pix2Pix | 1.106 | 0.088 | 0.147 | 3.193 | 23.940 | 0.823 |
| DeepFillv2 | **0.609** | 0.053 | 0.130 | 2.771 | 27.370 | 0.816 |
| SSDD-GAN (proposed) | 0.610 | **0.040** | **0.093** | **2.296** | **28.896** | **0.878** |

Table 1: **Quantitative Performance.** Comparison among various methods.

struction fidelity. Furthermore, Figure 4(d-f) highlights the effectiveness of our approach in handling varying mask sizes, showing that our method maintains higher accuracy and produces more reliable results even as the missing regions increase. These findings underscore the adaptability and generalization capability of our method compared to existing techniques. Figure 5 shows the randomly selected results of completing the missing region

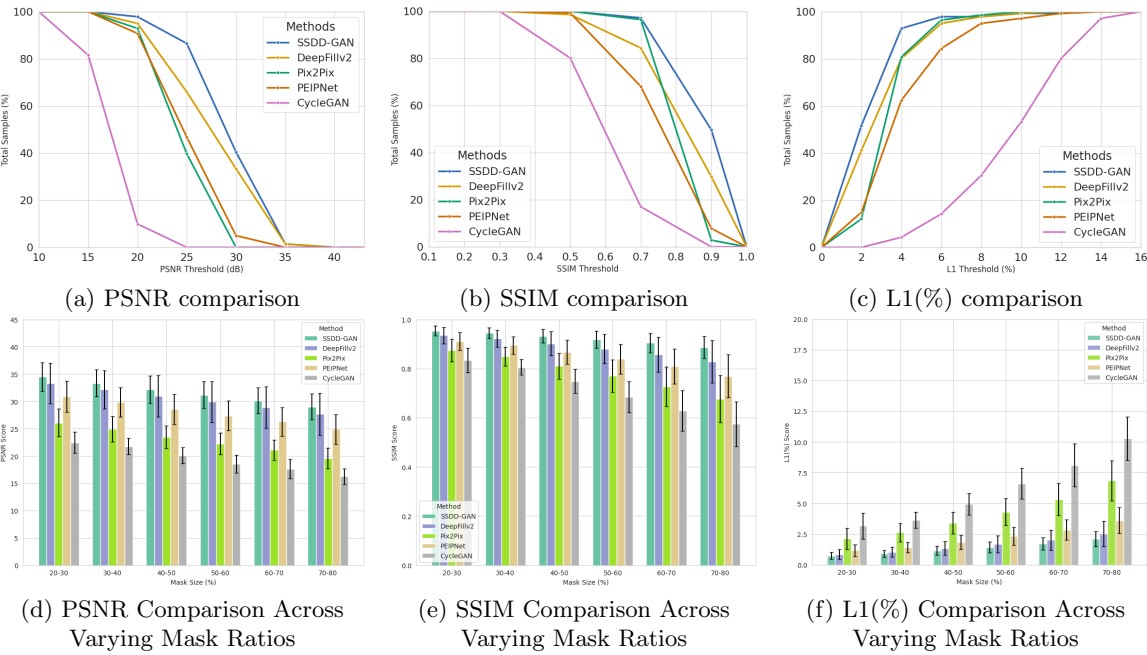

(a) PSNR comparison     (b) SSIM comparison     (c) L1(%) comparison

(d) PSNR Comparison Across Varying Mask Ratios     (e) SSIM Comparison Across Varying Mask Ratios     (f) L1(%) Comparison Across Varying Mask Ratios

Figure 4: **Performance Comparisons**. The experiments evaluate overall performance (**top row**) as well as performance across varying mask ratios (**bottom row**).

by our proposed method when compared with different models.

### 3.2. Ablation Study

Figure 6 illustrates the effects of sweeping across the number of diffusion steps $T$ on the L1(%), PSNR, and SSIM metrics. From the plots, it shows that increasing the diffusion steps generally leads to improved performance by an interval of 200. Specifically, the PSNR metric has a notable increase, highlighting enhanced image quality as the number of diffusion steps $T$ increases, with performance peaking within the 700 to 900 range. Similarly, SSIM

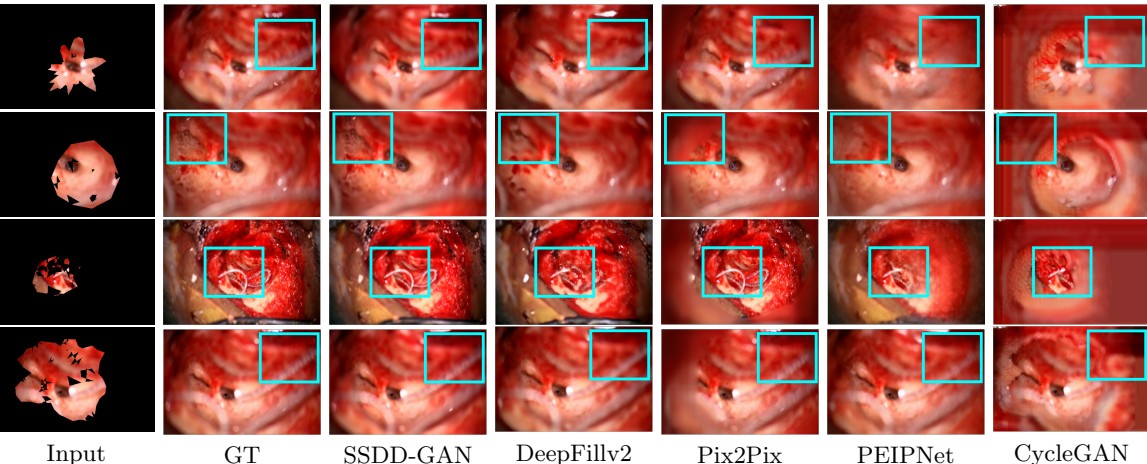

| Input | GT | SSDD-GAN | DeepFillv2 | Pix2Pix | PEIPNet | CycleGAN |

Figure 5: **Qualitative Comparisons**. Visualizations of completing missing regions using various methods. Details regarding to the noticeable improvement are highlighted in cyan bounding boxes.

metrics indicates a gradual increase in structural similarity with higher diffusion steps, plateauing at the 700-900 steps interval. Moreover, the L1(%) error consistently decreases with increasing diffusion steps, achieving optimal (lowest) values within the 700-900 range. The low performance observed in the range of 100 to 300 diffusion steps is likely due to an insufficient number of steps for noise addition during the diffusion process, and this drawback negatively impacts inference quality. For a complete comparison, we include the full range from 0 to 1000 to evaluate against the original configuration proposed in (Ho et al., 2020). In summary, these results demonstrate that the optimal performance across all metrics is achieved when the number of diffusion steps $T$ ranges from 700 to 900.

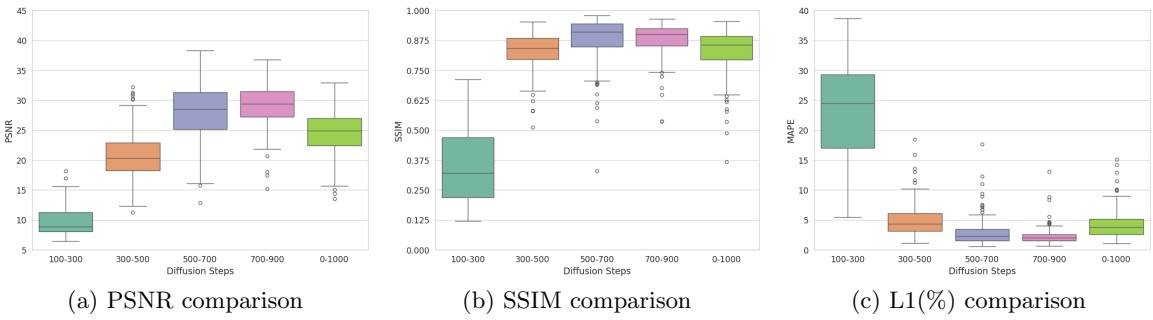

| (a) PSNR comparison | (b) SSIM comparison | (c) L1(%) comparison |

Figure 6: **Ablation Study**. Analyze the impact of varying the number of $T$.

### 3.3. Zero-shot Synthesis using the Synthetic Postmastoidectomy Dataset

Finally, Figure 7 shows the surgical scene completion results of missing regions in the synthetic postmastoidectomy dataset via the zero-shot approach. For comparison, we selected the closest real surgical frames to evaluate the quality of the synthetic surgical scenes. By

leveraging the precise camera pose information inherently generated within the synthetic postmastoidectomy dataset, our proposed method can fill the missing surgical scene that aligns well with the synthetic postmastoidectomy surface. This capability not only improves the realism of the synthetic surgical scenes but also represents a step forward in the surgical navigation field, with substantial potential to benefit a wide range of downstream tasks, including 3D scene understanding and anatomical structure tracking. We evaluate the com-

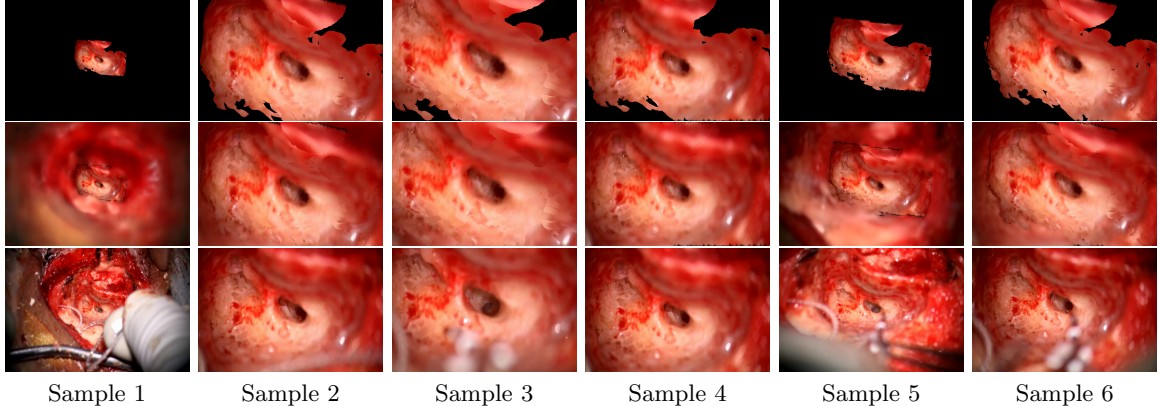

| Sample 1 | Sample 2 | Sample 3 | Sample 4 | Sample 5 | Sample 6 |

Figure 7: **Surgical Scene Synthesis**. Results of reconstructing the complete surgical field using the synthetic postmastoidectomy dataset. The first row shows original data, the second row presents the completed surgical scenes, and the final row displays the closest corresponding real surgical scenes.

putational efficiency of SSDD-GAN by measuring its inference speed. Our model achieves an inference rate of roughly 7 frames per second (~143 ms per frame) with approximately 35 million parameters on an NVIDIA GeForce RTX 4090 GPU.

## 4. Conclusion

The proposed method effectively completes missing regions in complex and cluttered surgical scenes, addressing challenges such as irregular geometries and occlusions introduced by the random masking technique. The use of self-supervised learning makes our method highly adaptable and generalizable to other surgical domains. Furthermore, the fully synthetic postmastoidectomy scenes provide precise camera pose information for each synthetic microscopy surgical view, paving the way for future advancements in the field of image-guided cochlear implant surgery. One limitation of the proposed method is its suboptimal performance when dealing with large missing regions in an image. This limitation arises from the difficulty of restoring fine details and textures in large missing areas using small known regions, a challenge prevalent in surgical datasets with intricate anatomical structures and complex textures. Future work could explore methods to address this limitation, and leverage these synthetic complete surgical scene multi-views to develop methods for intraoperative navigation of anatomical structures and accurate surgical tool tracking, providing better surgical guidance and potentially improving surgical precision and outcomes.

## Acknowledgments

This work was supported in part by grants R01DC014037 and R01DC008408 from the NIDCD. This work is solely the responsibility of the authors and does not necessarily reflect the views of this institute.

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

# Appendix A.  Qualitative Results of SSDD-GAN

Figure 8 demonstrates the effectiveness of the proposed framework in restoring various missing surgical scenes across different mask ratios.

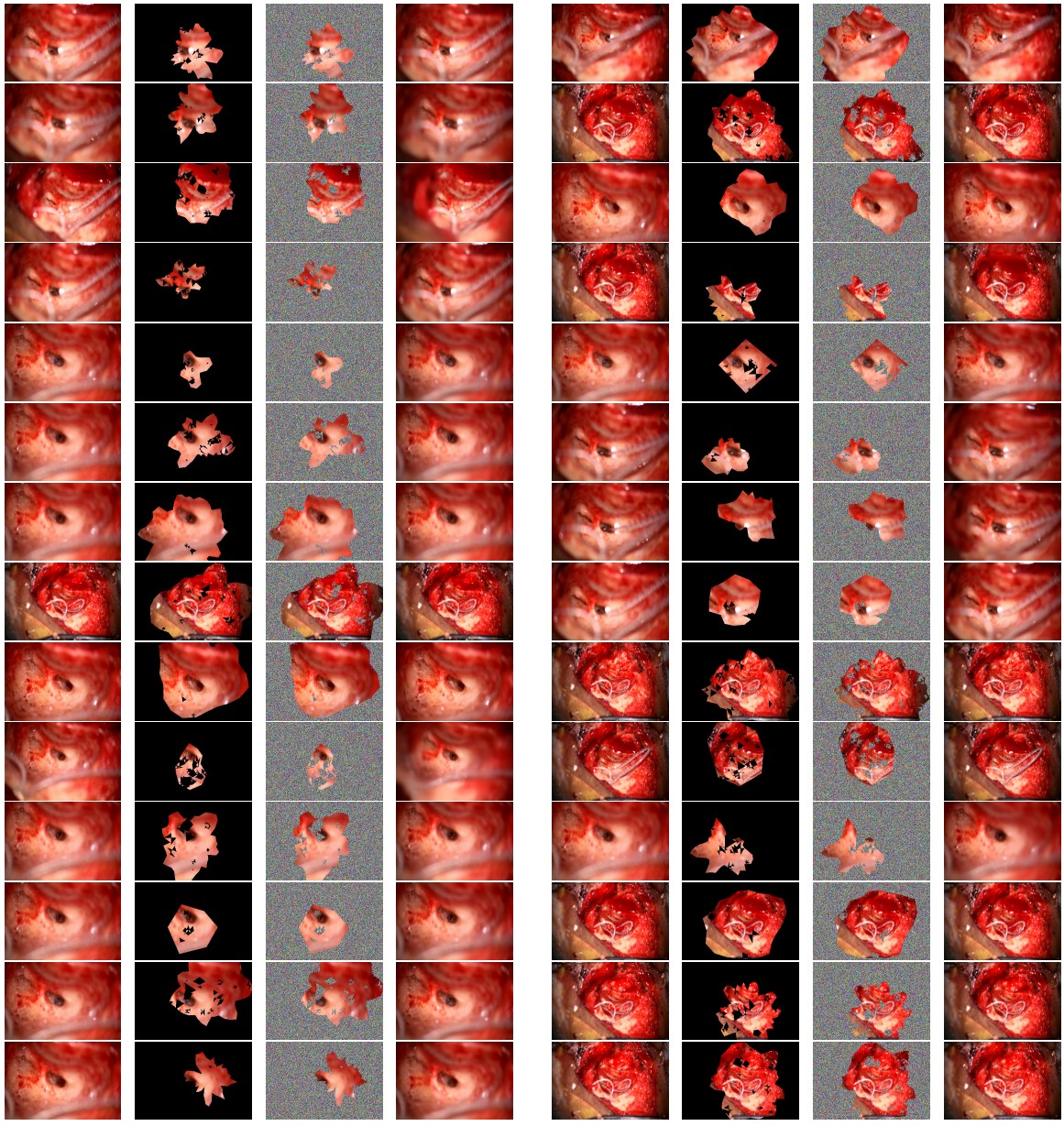

Figure 8: **Qualitative Performance Evaluation**. (a) Original Image. (b) Masked Image. (c) Diffused Image. (d) Reconstructed Image.

