# OpenReview forum: "SSDD-GAN: Single-Step Denoising Diffusion GAN for Cochlear Implant Surgical Scene Completion"
_MIDL.io/2025/Conference — MIDL 2025 Poster_

### Official Review · Reviewer_7jNa · 2025-02-21

**Confidence:** 2
**Preliminary Rating:** 4
**Final Rating:** 4

**Summary:**

The paper presents SSDD-GAN (Single-Step Denoising Diffusion GAN), a novel approach for surgical scene completion in cochlear implant procedures. It integrates denoising diffusion probabilistic models (DDPMs) with generative adversarial networks (GANs) to enhance image inpainting and outpainting. Unlike traditional diffusion models that require multiple steps for denoising, SSDD-GAN employs a single-step denoising process, reducing computational costs while improving realism via adversarial learning. The model is trained using self-supervised learning on real surgical datasets and applied in a zero-shot manner to synthetic postmastoidectomy datasets, enabling realistic surgical scene synthesis without explicit ground-truth labels.

**Strengths:**

1. The paper proposes SSDD-GAN, which effectively integrates denoising diffusion probabilistic models (DDPMs) with GAN-based adversarial optimization, achieving high-quality surgical scene completion with improved realism.

2. The model is trained using self-supervised learning on real surgical datasets, eliminating the need for manually annotated labels and improving its ability to generalize across diverse surgical environments.

3. SSDD-GAN is directly applied to synthetic postmastoidectomy datasets without additional fine-tuning, demonstrating strong zero-shot generalization capabilities for surgical scene completion.

4. The proposed method outperforms CycleGAN, Pix2Pix, DeepFillv2, and PEIPNet across multiple metrics, including SSIM, PSNR, and FID, achieving more accurate and realistic reconstructions.

**Weaknesses:**

1. While the paper claims computational efficiency as a motivation for single-step denoising, it lacks a thorough comparative analysis against multi-step diffusion models to quantify the trade-off between speed and quality.

2. The self-supervised learning approach relies on real surgical datasets, but the paper does not discuss data scarcity issues or domain adaptation strategies, which could limit applicability to new surgical settings.

3. The paper applies SSDD-GAN to a synthetic dataset in a zero-shot manner but does not compare its performance to models explicitly trained on the synthetic dataset, leaving uncertainty about the method’s adaptability.

4. The proposed method struggles when large portions of the surgical scene are missing, yet the paper does not explore alternative strategies, such as multi-scale context modeling, to mitigate this limitation.

5. The paper does not isolate the contributions of the diffusion process, adversarial training, and self-supervised learning in its experimental analysis, making it unclear which components drive the performance improvements.

**Detailed Comments:**

Please refer to weaknesses and strengths

**Justification Of The Final Rating:**

The rebuttal addressed my concerns. The paper introduces an innovative framework focused on the inpainting task for surgical scenes, addressing a meaningful problem in this domain. Therefore, I have decided to assign a **weak accept** score.

**Justification Of The Preliminary Rating:**

The paper presents a novel and efficient approach for surgical scene completion with strong results, but lacks sufficient justification for single-step denoising, robustness to large missing regions, and comprehensive ablation studies.

**Questions To Address In The Rebuttal:**

Please refer to weaknesses and strengths

---

> ### Author Response · Authors · 2025-03-07
>
> Thank you very much for the valuable comments and the overall support for our work.
>
> In the paper, we chose a single-step denoising diffusion approach to improve inference efficiency while still achieving high-quality results. Traditional diffusion models require iterative denoising over tens or even hundreds of steps, which leads to slow inference and high computation costs. SSDD-GAN performs the denoising procedure in one step and eliminates the lengthy sampling process. This advantage is significant by mapping noise to data in a single forward pass, we avoid the long runtime of many diffusion iterations​. The trade-off is that multi-step diffusion can incrementally refine details, potentially yielding higher fidelity in reconstructed images. However, our quantitative and qualitative results show that the quality loss is minimal with SSDD in general. Thanks to the integration of a GAN discriminator, our single-step model retains fine details and realism compared with prior arts. In practice, we also tried the multi-step diffusion models, and found the multi-step diffusion models are difficult to incorporate with a GAN discriminator due to the high computational cost in the back-propagation process and our current setup does not support us to perform such computational intensive tasks. However, it could be a valuable direction to explore the trade-offs in following works.
>
> We appreciate the reviewer’s point. Our method leverages a real surgical dataset (932 frames from a cochlear implant surgery) for self-supervised training, which does not contain many images in the setting. A key advantage of our approach is that it is self-supervised, meaning it does not require any manual annotations from the real data. This eliminates the burden of data collection in new domains (as we only need to record a short video). Any unlabeled raw images can be used to train on their specific domain. The use of self-supervised learning makes our method highly adaptable and generalizable to other surgical contexts as well. We only need a relatively small number of raw images from the target surgical setting, and we can apply the same random masking generation procedure to guide the model learning the new domain’s features and distributions. We exclude the pseudocode that discusses the automatic mask generations in current submission, but we have included them in the revised version (shown in Appendix C) for this paper.  For extending SSDD-GAN to a new surgical setting, we can think of several domain adaptation strategies for now, such as fine-tuning with few samples and directly re-train the model with other domain images. The extensive evaluation and comparison of different surgical domains (e.g. endoscopy images) will be included in the future work.
>
> This is a valid point, and we will clarify this aspect further. Our work was motivated by the scenario where the synthetic postmastoidectomy dataset lacks ground-truth for the surrounding scene, which is why we trained on a real dataset and applied the model to the synthetic images via zero-shot learning. We did not compare to a model trained purely on synthetic data since the synthetic dataset by itself does not provide complete surgical scenes as ground truth for training a conventional inpainting model. In the synthesized postmastoidectomy dataset, all the novel camera views are obtained from a single microscopy image by adjusting the camera extrinsic matrix. Thus, they are partially completed as they only focus on the postmastoidectomy surface, but fail to synthesize any surgical surroundings and overall surgical environment. This paper addresses the prior limitation by training on the real surgical dataset and applying the model directly to the synthesized postmastoidectomy dataset via zero-shot to complete the full surgical scene. We will add more details regarding the generation of the synthesized postmastoidectomy dataset in the introduction to help address the concern.
>
> Our method’s performance is suboptimal in large missing regions in the synthetic dataset; however, in real images, the model handles the missing large context better than prior arts, as evidenced by Figure 4 and Appendix A. Also, we conduct an experiment that tests our model’s performance with varying mask ratios from 20% to 80% (shown in the bottom row of Figure 4), which supports our claim of generating better images on the real image dataset compared with previous methods. For directly applying the proposed method to the synthesized postmastoidectomy dataset via zero-shot, possible solutions to mitigate the performance issue could be the multi-scale context modeling, and we aim to explore these methods in future work. Thank the reviewer for suggesting a valuable point.
>
> We agree that including a more detailed study to discuss SSDD-GAN would strengthen the work, and we aim to provide that in the revised version.

---

### Official Review · Reviewer_pyFt · 2025-02-23

**Confidence:** 4
**Preliminary Rating:** 4
**Recommendation:** Poster
**Final Rating:** 4

**Summary:**

The authors propose SSDD-GAN to reconstruct the complete post-mastoidectomy surgical scene by training a neural network on
real surgical datasets.

The introduction and the motivation is written well.

Combining the diffusion process with an adversarial learning setting is interesting.

**Strengths:**

The paper motivates a self-supervised learning approach using diffusion models and GANs.

The approach is patient-specific, paving the way for surgical scene understanding via
intraoperative registration between preoperative CT scans and the corresponding surgical
scene. In addition, the approach supports tool tracking and anatomical navigation, offering the potential for improving preoperative planning and intraoperative guidance for cochlear implant surgery.

**Weaknesses:**

It is unclear why the authors call their method a single-step denoising as a diffusion model. The spirit of diffusion models is to iteratively and progressively denoise an image in the reverse step. I'm asking this because VAEs are single-step generative models reconstructing from standard normal distribution N(0, I); attaching an adversarial learning setting is very similar to VAE-GAN to prevent the blur introduced in vanilla VAEs. By making diffusion models a single step and adding a discriminator, the authors aim to build a VAE-GAN. If not, an explanation at the beginning of Section 2 must be provided.

Adding a discriminator to the single-step denoising routine can lead to satisfactory results since the discriminator provides an additional adversarial learning signal that helps the diffusion model refine the predicted noises -----------> Do we have a citation for this? Else how?

Eq. 1 has three parts; the first is redundant and can be removed.

The approach assumes that the binary mask is present.

Note that δxt
is equivalent to δx0 in our setting. -------> This is an ambiguous statement

In figure 2, it is unclear why the t+1 step is used within the single-step denoising block.

Please indicate by arrows in Fig.4 where the recovery is better than other approaches.

As time is claimed as a critical component, the computation cost analysis is missing.

**Detailed Comments:**

Section 2: Does the term "targeted region" refer to those in black in the binary mask? Is it an automatic segmentation?

How is the binary mask computed?

Please elaborate on Figure 2 for readability. For example, it is unclear where δx0 is in Figure 1 or 2 in the forward process.

Is Eq 4 obtained by refactoring Eq 3?

**Justification Of The Final Rating:**

Most of my review comments related to differentiating the VAE-GAN with the single-step denoising approach, notations, the concept diagram, removing redundant equations, and improving the clarity of the visual results are addressed. Hence I would like to retain my rating.

**Justification Of The Preliminary Rating:**

I have given weak acceptance because of the generative model direction that the authors have taken building upon their previous work, taking into consideration the merits and demerits of the baseline.

**Questions To Address In The Rebuttal:**

Most of my questions are presented in the previous section. I request the authors to refer to those questions.

---

> ### Author Response · Authors · 2025-03-07
>
> Thank you very much for the thorough and constructive feedback. Also, the support for our work!
>
> **For the weakness:** We appreciate the opportunity to clarify the intention of why we use SSDD-GAN instead of VAE-GAN in the paper. We could consider diffusion models to be a special case of VAEs proposed by [1]. SSDD-GAN stands for the "single-step denoising diffusion" because it simplifies the iterative reverse diffusion process used in traditional diffusion models (DDPMs) into just one direct step from noise to reconstruction. Unlike typical diffusion methods requiring multiple steps, our model learns to map noisy inputs directly to completed images. Although superficially similar to VAE-GAN architectures, our method specifically maintains the diffusion formulation: training explicitly with progressive noise addition (no-trainable parameters in the encoder) and subsequent denoising objectives. Unlike VAEs, which encode to a latent vector first, our approach directly predicts the original image conditioned on noise-added input, preserving diffusion-based conditional generation while benefiting from adversarial components provided by GAN. Overall, we consider the name SSDD-GAN to be better than VAE-GAN to describe the framework. We aim to emphasize these distinctions explicitly in Section 2 of the revised paper.
>
> We understand the request for a citation regarding the integration of the discriminator into the single-step diffusion framework. While our idea of integrating adversarial training into diffusion-inspired methods for image inpainting tasks is novel, conceptually similar combinations (but still very different) have been explored in prior works such as Diffusion-GAN [2]. We will add such references in our revised manuscript to provide a clear context.
>
> We thank the reviewer for pointing out the redundancy in Eq. 1 and agree that the first part of Eq. 1 may appear unnecessary. We will remove that part in the revised paper. We recognize that our notation "Note that δxₜ is equivalent to δx₀ in our setting" might be unclear. We intended to deliver the point that the mask region remains unchanged from the original image (x₀) throughout the diffusion process. We will rephrase this sentence more clearly as "δxₜ refers to masked regions that are identical to the corresponding regions in the original image x₀ throughout the diffusion process." In Figure 2, even though we are performing the one-step diffusion process, the images being denoised still correspond to the original noise-injected images at the specific timestep t+1. We intended to highlight the correlation between the denoised outputs and their corresponding t noise-added images, thereby providing a more direct comparison with the iterative process shown in the original DDPM paper.
> Thank you for suggesting a valuable point regarding Figure 4, and we will add cyan bounding boxes to indicate where the recovery is better than other approaches per request. We have added a computational cost analysis in our paper to demonstrate the efficiency of using our single-step denoising network.
>
> **For the detailed comments:** Yes, the "targeted region" corresponds to the black area in the binary mask for Figure 1, while in Figure 2, it refers to the white area in the inverted binary mask. For our training approach, these masks are automatically generated through random polygonal shapes simulating realistic missing regions in the synthesized postmastoidectomy dataset, thus requiring no manual/automatic segmentation. Due to the page limitations, we exclude the pseudocode that better demonstrates the mask generation process. We have included the pseudocode to the revised version, as shown in the Appendix C. We have further elaborated on Figure 2 more thoroughly in the revised paper. Eq 4 was obtained not directly from refactoring Eq 3, since we did have some modifications based on the actual code implementation that take into consideration the masked region (which is not altered in our approach).
>
> [1] Sohl-Dickstein, J., Weiss, E. A., Maheswaranathan, N., & Ganguli, S. (2015). Deep Unsupervised Learning using Nonequilibrium Thermodynamics. arXiv [Cs.LG]. Retrieved from http://arxiv.org/abs/1503.03585
>
> [2] Wang, Z., Zheng, H., He, P., Chen, W., & Zhou, M. (2023). Diffusion-GAN: Training GANs with Diffusion. arXiv [Cs.LG]. Retrieved from http://arxiv.org/abs/2206.02262

---

### Official Review · Reviewer_e9yy · 2025-02-24

**Confidence:** 4
**Preliminary Rating:** 3
**Recommendation:** Poster
**Final Rating:** 3

**Summary:**

This paper proposes an efficient method for completing surgical scenes in synthetic datasets. It utilizes self-supervised learning to train a one-shot denoising GAN, leveraging adversarial training and diffusion models. The proposed approach enables full surgical field visualization with precise camera pose information, enhancing the usability of synthetic datasets. The method demonstrates significant improvements in dataset performance.

**Strengths:**

+ This paper introduces an innovative inpainting method for surgical scenes using a self-supervised learning approach.
+ The model leverages the strengths of diffusion models and adversarial training to address the limitations of previous methods, enabling the reconstruction of full surgical scenes.
+ The paper is well-written and effectively organized.

**Weaknesses:**

- The work utilizes a well-established architecture from recent deep learning papers applied to a medical-related dataset, which reduces the novelty in the architectural design.
- Additionally, the evaluation is limited due to the lack of comparisons with recently published works, making the paper less compelling.
- Including comparisons with prior studies specifically targeting this task and discussing the strengths and weaknesses of each method would enhance the presentation of the proposed algorithm's effectiveness.
- Furthermore, more ablation studies are required to thoroughly demonstrate the contributions and effectiveness of the GAN and diffusion components.

**Detailed Comments:**

1. The paper should include comparisons with additional relevant papers to better demonstrate the effectiveness of the proposed method.
2. Expanding the explanation of Figure 5 is necessary to provide clarity, along with including visualizations for all metrics discussed in the paper.

**Justification Of The Final Rating:**

Thank you to the authors for their detailed feedback. This paper introduces an innovative framework focused on the inpainting task of surgical scenes, addressing a significant topic in this area. However, similar work, such as Diffusion-GAN, reduces the novelty of the proposed approach. While the rebuttal addressed some of my concerns, I remain uncertain whether the final version will incorporate all the suggested revisions. Therefore, I maintain my final rating as borderline.

**Justification Of The Preliminary Rating:**

This paper introduces an innovative framework focused on the inpainting task of surgical scenes, addressing a significant topic in this area. While the proposed method combines architectures effectively, the lack of substantial architectural innovation and insufficient comparisons with recent state-of-the-art (SOTA) methods detract from the paper’s overall contribution.
I strongly recommend including additional results and analyses to substantiate the proposed method’s effectiveness and versatility. Expanding evaluations to other datasets and incorporating comparisons with recent SOTA methods would significantly enhance the paper’s robustness and impact. Thus, my final rating is Borderline.

**Questions To Address In The Rebuttal:**

MaskVAE could be an alternative to the current approach, why the proposed method is chosen?
Video inpainting introduces challenges such as temporal coherence and frame continuity, which the current method would need to address for video impainting?

---

> ### Author Response · Authors · 2025-03-02
>
> Thank you very much for the thoughtful feedback and valuable suggestions.
>
> **For the weakness:** SSDD-GAN as a _Validation_ _or_ _Application_ focused paper leverages well-established components, but our key novelty lies in how these components combine into a single-step denoising process and be able to only reconstruct the targeted masked area. Traditional diffusion models generate images through many iterative denoising steps, and GAN-based inpainting models typically operate in one step but without the diffusion process; thus leading to lower performance than the diffusion models in general. We combine diffusion and adversarial training in a one-step pipeline, meaning the model takes a noisy image and produces a completed scene in a single forward pass. This is unlike prior works which either run costly using the multi-step denoising processes​ or use GANs alone without the nice diffusion’s noise injection. To our best knowledge, no previous work has combined a diffusion model and GAN in a single-step inpainting setting for surgical images, making our framework distinctive. In summary, SSDD-GAN’s novelty is in merging and extending prior arts into a unique framework for injecting Gaussian noise only into masked regions during training and preserving unmasked regions​. Our strategy results in more efficient training and fast inference time. Also include a 6% SSIM improvement over prior methods by capturing both coarse content and fine details together. We'd like to thank the reviewer for encouraging us to highlight these innovations more clearly.
>
> In our paper, we compared SSDD-GAN against representative generative models used for image completion: for example, CycleGAN and Pix2Pix were included as baseline methods​. These classic works were chosen because, at the time of submission, no prior work specifically tackled cochlear implant surgical scene completion, so we adapted well-known image-to-image translation and inpainting frameworks for a baseline. At the same time, some recent work was chosen as well, such as the well-known deepfillv2 and most recent method PEIPNet. SSDD-GAN outperformed these methods by a large margin in most metrics. We understand that more recent or specialized approaches (e.g. diffusion-based inpainting models or Masked VAE frameworks) could be relevant for comparison. Many of these, however, were not straightforward to apply in our scenario. For instance, some SOTA diffusion inpainting methods (such as general Stable Diffusion-based inpainting or RePaint) are trained on natural images and require text prompts or very large training sets, making direct comparison on our surgical dataset difficult without significant adaptation. Also, majority diffusion-based models can completely alter the known content and produce some images that are entirely new, which does not fit for our purpose. Similarly, purely VAE-based with GAN component inpainting methods tend to produce blurrier results and were less likely to handle the fine anatomical details needed. Due to dataset limitations and domain differences, we focused on baselines that we are able to train under our current available setups on this specific task. We appreciate the reviewer’s understanding and will treat this evaluation as an opportunity for future work.
>
> Thank you for suggesting a valuable point. We agree that more detailed study is needed in our paper. In the current submission, page restrictions limited our discussion, but we have included more details in our revised paper.
>
> **For the detailed comments:** Thank you for bringing concerns regarding Figure 5. We have revised the paper to improve clarity in our visualization. Figure 5 was intended to summarize our model’s performance across different diffusion steps and explain why we pick the diffusion steps between 700 and 900. In the initial submission, the description was too brief, largely due to the page limitation. We have provided a more detailed explanation in the revised paper.
>
> **For the questions:** The reviewer raises a good point about considering MaskVAE-based approaches. We chose not to use a MaskVAE primarily because our goal was to maximize output fidelity and realism, and diffusion/GAN models have recently outperformed VAE-based models in these aspects. In a complex surgical scene, we found it crucial to capture fine details (edges of the malleus of ossicles, bloody holes location, etc.), where GANs and diffusion models perform well. In our previous experiments with standard autoencoder baselines, the outputs lacked important texture/details for the surgical scene. We chose not explore video-specific strategies in our work, as our focus was on static image scene completion, but we see the importance for surgical video applications. We appreciate the reviewer bringing up this point, and while it’s beyond the current scope, we are excited to explore it in subsequent work.

---

### Author Rebuttal · Authors · 2025-03-07

**Rebuttal:**

We have attached an updated manuscript with the revised content highlighted.

**Supporting Material:**

/attachment/b4ef9e9b93c135aec5b4c66fc0fc3d130347e726.pdf

---

### Meta-Review · Area_Chair_xLvf · 2025-03-21

**Recommendation:** Accept (Poster)
**Confidence:** 4

**Metareview:**

The reviewers' overall opinion leans positive after the authors' rebuttal. The authors are strongly encouraged to incorporate all suggested revisions from the rebuttal into the final version.